# Polyclonal B cells acquire LCMV antigens in a GP1-dependent manner

Gabriel Chamberlain[1], Guillaume L. Lopez[1], Léa Bourguignon[1], Xavier Laulhé[1], Yasmine Adda-Bouchard[1], Tania Charpentier[1], Rebekah Honce[2], Jason W. Botten[2], Alain Lamarre[1]*

1 Immunovirology Laboratory, Institut national de la recherche scientifique (INRS), Centre Armand-Frappier Santé Biotechnologie, Laval, Québec, Canada, 2 Department of Medicine, Division of Pulmonary Disease and Critical Care Medicine, Robert Larner,M.D. College of Medicine, University of Vermont, Burlington, Vermont, United States of America

* alain.lamarre@inrs.ca

## Abstract

The polyclonal, T-dependent nature of hypergammaglobulinemia during murine infection with LCMV is well defined, however the mechanism by which polyclonal B cells acquire antigens for presentation remains unknown. Here we use LCMV-specific CD4 + transgenic T cells to explore several hypotheses for B cell antigen uptake. We found that antigens produced by cells infected with LCMV *in vitro* are available to polyclonal B cells and their presentation to CD4+ T cells enabled robust co-activation. The *in vitro* nature of our model demonstrates that *in vivo* factors such as cytokine milieu and antigen release by NK/CD8+ are not required. We show that non-replicative UVC-irradiated LCMV enables antigen access to B cells, thus productive infection of B cells is not required for antigen acquisition. B cells loaded with LCMV antigens *in vitro* can efficiently present these antigens to CD4+ T cells in LCMV-infected mice, thereby validating our model *in vivo*. Using transmission electron microscopy we identified LCMV-GP bearing particles of various sizes including <50nm, a size accessible to B cells via pinocytosis. The particles morphologically resemble exosomes or viral particles (infective or defective interfering). We show that polyclonal B cell access to Ag is maintained when exosome release or viral defective interfering particle release is inhibited. Finally, we used a monovalent Fab derived from the LCMV-neutralizing antibody KL25 to show that access to the viral GP1 protein is important in order for polyclonal B cells to efficiently acquire LCMV antigen for presentation, suggesting the presence of a GP1 receptor on B cells.

## Author summary

Murine lymphocytic choriomeningitis virus (LCMV) infection is a powerful model that has enabled researchers to make countless important immunological

**Data availability statement:** All relevant data are within the manuscript and its Supporting Information files.

**Funding:** This work was supported by operating grants from the Canadian Institutes of Health Research (CIHR) to AL (PJT-159525) and National Institutes of Health (NIH) to JWB (R01 AI171408 and R21 AI154198) and RH (T32HL076122). AL holds the Jeanne and J.-Louis Lévesque Research Chair in Immunovirology from the J.-Louis Lévesque Foundation. The funders had no role in study design, data collection and analysis, decision to publish, or preparation of the manuscript.

**Competing interests:** The authors have declared that no competing interests exist.

discoveries. LCMV offers a convenient and well-characterized model to study acute versus persistent viral infections as well as many of the same immune dysregulations seen in human chronic viral infections. Specifically, LCMV causes T-dependent hypergammaglobulinemia which is more pronounced during chronic infections. In this paper, we use LCMV to study how polyclonal B cells acquire antigens to present to cognate helper T cells, a necessary event in establishing hypergammaglobulinemia. Our aim is that by solving this unanswered question with the LCMV model we may glean valuable insights that could lead to advances in combatting viral infections in humans. In this study we use various techniques including the direct assaying of antigen presentation between polyclonal B cells and LCMV specific CD4 T cells to exclude many possible hypotheses for B cell antigen acquisition. Collectively, the results of our study provides the first evidence for a receptor on polyclonal B cells that recognizes LCMV GP1.

## Introduction

T-dependent polyclonal hypergammaglobulinemia (HGG) is an important consequence of many chronic viral infections, including with human immunodeficiency virus (HIV) and hepatitis C virus (HCV). Murine infection with LCMV also leads to HGG and offers a convenient model for the study of this phenomenon. Various strains of LCMV exist, infection of C57BL/6 mice with the Armstrong (ARM) or, WE strains lead to acute infection while infecting the same mice with the Clone 13 (Cl13) variant of the Armstrong strain, or the LCMV-Docile strain results in chronic infection. While chronic infection typically results in the greatest levels of HGG, the phenomenon is also present during acute infection [1].

The polyclonal, T-dependent and BCR-independent nature of HGG during murine infection with the Old World arenavirus LCMV is well defined [1]. However, by what mechanism polyclonal B cells can acquire LCMV antigens (Ags) for presentation to cognate helper T cells remains unknown.

Interestingly, murine infection with recombinant LCMV Cl13 lacking the glycoprotein 1 (GP1) MHC class II epitope GP61–80 (GP61) does not induce HGG [2], possibly due to the preponderance of helper T cells specific for this immunodominant epitope being a requirement. To understand how HGG is initiated and maintained during LCMV infection, it is therefore critical to understand how B cells acquire and present LCMV Ags (predominantly GP61) to specific T cells.

Several hypotheses exist for antigen acquisition by polyclonal B cells during LCMV infection ranging from receptor-independent pinocytosis to, controversially, the direct infection of B cells by LCMV. One mucosal study suggests a small but biologically relevant portion of B cells may be infected in mice rectally inoculated with LCMV [3]. However, in most other cases, direct infection of B cells by LCMV remains unlikely as neither viral RNA nor protein is easily detectable in B cells from mice intravenously or peritoneally inoculated with LCMV [1,4,5].

It has previously been shown that VSV specific µMT B cells are capable of taking up and presenting recombinant LCMV NP to LCMV NP-specific VE8 T cell hybridomas [1]. This suggests that polyclonal B cells may acquire LCMV Ags in the same way they acquire recombinant peptides *in vitro*; via receptor-independent pinocytosis. Furthermore, it has been demonstrated that polyclonal B cells that are loaded with recombinant LCMV peptides in a BCR-independent manner *in vitro* and subsequently transferred to LCMV infected mice will proliferate and produce polyclonal antibodies thus driving HGG [6]. For pinocytosis to occur there must be sufficient concentrations of small soluble viral antigens present and available for passive capture in the fluid surrounding B cells. The upper size limit of molecules able to enter B cells through pinocytosis has not been explicitly studied however recent research focusing on the effects of nanoplastics shows that latex beads up to 50nm in diameter are internalized by (and are detrimental to) both B and T cells [7]. There are many possible mechanisms by which, in theory, LCMV antigens could be present in a sufficiently small and soluble format. The viability of many viruses is known to decline as a function of temperature over time, it might be possible that virions that do not find host cells quickly at physiological temperatures could be subject to degradation into smaller fragments more amenable to pinocytosis. HIV-I, a virus that similarly induces HGG, is known to shed glycoprotein peptides at various temperatures, including 37°C, over time [8]. Another potential source of soluble LCMV GP could be shedding from the membranes of infected cells. Ectodomain shedding of viral glycoproteins on infected cell membranes by proteolytic enzymes known as sheddases is a common phenomenon among many cell types and viral infections [9–11]. It has been observed that humans infected with the Lassa virus (LASV), another arenavirus, have detectable levels of soluble LASV GP in their blood and it is believed these are the result of membrane-bound GP lost to shedding before viral budding is complete [12]. Additionally, LCMV is known to produce defective interfering (DI) particles, which can enter permissive host cells and block infectious particles from completing the viral life cycle [13]. It is possible that DI particles, exosomes released from infected cells, or cellular debris released from NK/CTL-mediated killing of infected cells might liberate sufficient concentrations of soluble Ags to make antigen acquisition through pinocytosis possible. Similarly, cells infected with LCMV continue to release exosomes which contain viral antigens and have an impact of CD8+T cell function [14]. Much of what is known regarding BCR-independent antigen presentation has been elucidated through experiments using small soluble antigens delivered *in vitro* via pinocytosis such as ovalbumin, hen-egg lysosome and, recombinant LCMV antigens such as GP and nucleoprotein (NP) [6,15,16]. Therefore, with regards to receptor-independent pinocytosis, the question is not *whether or not* B cells can present antigens via this mechanism, rather, *do B cells have access* to LCMV antigens of a permissible size?

In this study, we aimed to determine how polyclonal B cells gain access to the immunodominant LCMV antigen GP61. Here we show that polyclonal B cells can present GP61 derived from LCMV-infected cell supernatant to LCMV GP61-specific TCR transgenic CD4+T cells. Our model offers a convenient system to study the requisites of antigen acquisition devoid of many confounding factors such as the cytokine milieu or NK/CD8+T cell-mediated killing of infected cells. Additionally, supernatants from *in vitro*-infected cells can be assayed for the presence of LCMV antigens of various sizes at higher concentrations than those derived from infected mouse serum. We show that antigen acquisition by B cells is maintained when the virus is rendered non-replicative through UV irradiation, however uptake is significantly reduced by the addition of F(ab) from the LCMV neutralizing antibody KL25. Our findings suggest that polyclonal B cells may possess a receptor capable of recognizing LCMV GP1 but that productive intracellular replication with LCMV is not required to present sufficient amounts of antigen to receive T-cell help.

## Results

### Soluble GP is undetectable in LCMV-infected cell culture supernatant

Receptor-independent endocytosis of soluble, low molecular mass antigens has been proposed as a potential source of antigen acquisition for B cells driving HGG in LCMV [1]. The immunodominant MHC class II T cell epitope

GP61, necessary for driving HGG [2] is located on the GP1 molecule, therefore if GP shedding occurs during LCMV infection as it does with LASV infection [9,12], it would constitute a readily available source of antigen for polyclonal B cells.

To determine if infection with LCMV leads to GP shedding we used a similar method as the one described by Branco and colleagues [12]. Briefly, we collected LCMV-infected MC57G cell culture supernatant from various time points post-inoculation and probed for both GP and NP using monoclonal antibodies WEN3 and VL4 respectively. Timepoints where NP is detected would represent the presence of whole virus in the supernatant, while detection of GP-only would suggest GP shedding before the release of whole virion. Virus can be detected through plaque assay beginning at 16 hours post inoculation (p.i.) so we focused our investigation of GP shedding on the 12 – 20h p.i. window to capture the moment just before and after the beginning of viral release. We also included 24 and 48 hour timepoints when strong viral titers are easily detected with both plaque assay and dot blot assays. Our analysis did not detect any evidence of GP shedding using this temporal separation method (Fig 1A). Because it may be possible that soluble GP is released at the same time as whole virus, we performed a second analysis using mechanical separation. We passed the infected cell culture super-natant through a 100 kDa molecular weight cut off (MWCO) filter, LCMV GP1 monomer is a 44 kDa protein and if present would be detectable in the filtrate. We probed the < 100 kDa filtrate and >100 kDa retentate for the presence of GP1 using the WEN3 monoclonal antibody in both ELISA and dot blot assays. Neither NP nor GP1 were detected in the < 100 kDa filtrate using the dot blot (Fig 1B). The presence of GP1 was confirmed in the > 100 kDa retentate and the unfiltered viral supernatant but not in the < 100 kDa filtrate, which was not statistically significant when compared with uninfected cell supernatant (mock) (Fig 1C).

## LCMV-infected cells release antigens available to polyclonal B cells *in vitro*

To ascertain that our inability to detect GP shedding wasn't simply due to limitations in the sensitivity of our methods of immunodetection, we proceeded to test for soluble GP1 antigen using a T cell-B cell antigen presentation assay (TBAP). We used supernatant from LCMV-infected cell cultures and as before, passed these through a 100kDa filter keeping both retentate and filtrate. Polyclonal B cells were incubated with either 100kDa filtrate (100kDa fil) or retentate (100kDa ret) overnight. Successful B cell acquisition of LCMV antigen was determined by the B cells' ability to present antigen to GP1-specific CD4 + T cells isolated from TCR transgenic SMARTA-I mice. Antigen presentation was determined by measurement of SMARTA-I CD4 + T cell proliferation (CFSE loss), increased B cell expression CD86 and the presence of IL-2 and IL-6 in the co-culture supernatant (Fig 2A-E). In some of our assays we observed what we believe to be proliferation-independent CFSE loss, similar to what is described by others [17]. In assays where only a single CFSE division occurs in negative controls, the two highest CFSE+ populations are not included in the proliferation gate (Fig 2A-E). Compared to negative controls, we detected significant levels of IL-2 and IL-6 as well as CFSE loss in T cells and CD86 upregulation in B cells in conditions where B cells had been incubated with antigens from the > 100kDa reten-tate (Fig 2A-E). No significant changes in CFSE or CD86 were observed compared to negative controls in conditions where B cells had been incubated with antigens from the < 100kDa filtrate (Fig 2A-C). Similarly, IL-2 and IL-6 could not be detected in conditions where B cells had been given < 100kDa filtrate (Fig 2D and 2E). Importantly, no T cell proliferation was observed in the TBAP assay when B cells loaded with LCMV antigen were co-cultured with CD4 + T cells from wild-type mice (S1A Fig). This latter result demonstrates that T cell proliferation and B cell activation are specifically the result of antigen presentation to cognate T cells. Even though the filter sizes used should have allowed soluble GP1 to pass through, it is possible that other molecular interactions hindered this process. To ensure that our results reflected the nature of the antigen rather than unrecognized technical limitations, we repeated the assay using the filtrate and retentate obtained from the filtration (100 kDa) of recombinant LCMV GP. As anticipated, we found that while some of the antigen remained in the retentate, a larger portion was able to pass through the filter, as indicated by a higher proportion of prolif-erated cells in the filtrate condition (S1B Fig).

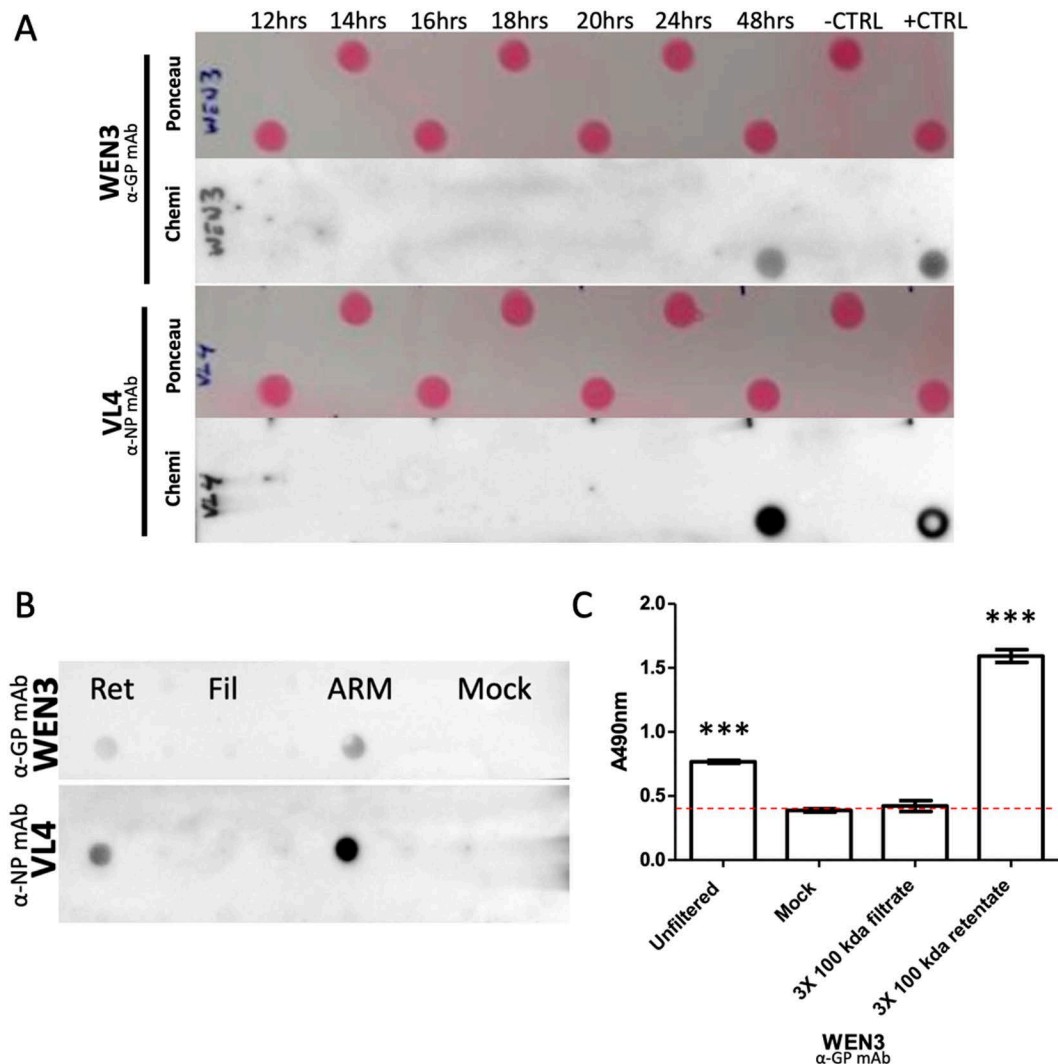

**Fig 1. Immunodetection of LCMV GP1 and NP from infected viral supernatant provide no evidence for GP shedding.** (A) PVDF membranes with dot blots from LCMV-infected cell supernatants collected at various timepoints labelled with ponceau (red dots) and then anti-LCMV GP antibody WEN3 or anti-LCMV NP antibody VL4 (black/grey dots). (B) PVDF membrane with dot blots of uninfected cell supernatant (Mock), 48h post inoculation LCMV-infected cell supernatant (ARM) and, the retentate (Ret) or filtrate (Fil) of LCMV-infected cell supernatant passed through a 100 kDa MWCO centrifugal filter. (C) Absorbance measurements at 490nm wavelength of unfiltered 48h post inoculation LCMV-infected cell supernatant (Unfiltered), uninfected cell supernatant (Mock) or the concentrated filtrate or retentate of 48h post inoculation LCMV-infected cell supernatant passed through a 100 kDa MWCO centrifugal filter probed with WEN3 using an ELISA, * denotes significance (P < 0.05) with respect to Mock in an unpaired student's T-test, the dotted line represents the background threshold.

We next considered the possibility that GP1 may be shed as a trimer, as it is displayed on virions. Therefore, we repeated the same methodology as described above instead using a larger 200kDa filter. Again, we detected no significant change in B cell expression of CD86, no T-cell proliferation or any IL-2 or IL-6 cytokine production in conditions where B cells were incubated with antigens from the < 200kDa filtrate (Fig 2A-E). These results indicate that GP1 available to B cells is present on protein molecules larger than 200kDa. We conclude that the source of GP1 acquired by polyclonal B cells does not match the expected sizes of GP1 monomers (44kDa) or trimers (132kDa) originating from GP shedding. Importantly, our results here indicate that our antigen-presentation assay is suitable

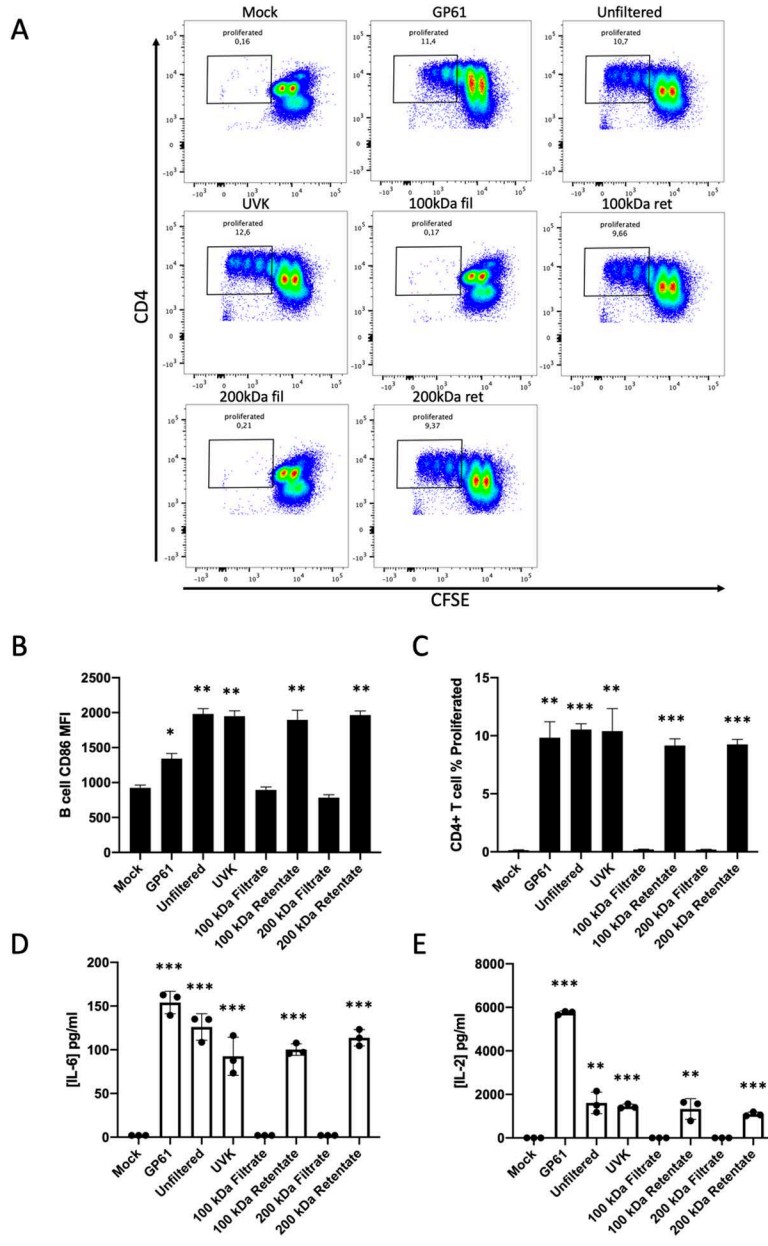

**Fig 2. LCMV-infected cells release antigens available to polyclonal B cells in vitro.** (A) FACS plots of CFSE labeled SMARTA-I CD4+T cells after 72 hours of co-culture with antigen loaded wild-type B cells, each plot shows data from one mouse, and each assay was independently repeated a minimum of three times ($n=3$). (B) CD86 expression on B cells measured by mean fluorescent intensity. (C) Percentage of SMARTA-I CD4+T cell proliferation measured by gating on CFSE low cells (example gate seen in panel A). (D) IL-6 concentration in pg/ml detected in 72 hour co-culture medium measured by ELISA. (E) IL-2 concentration in pg/ml detected in 72 hour co-culture medium measured by ELISA. In all bar graphs values are averages of three experiments ±SEM, * denotes significance ($P<0.05$) with respect to Mock in an unpaired student's T-test.

for detecting LCMV antigen available for acquisition by polyclonal B cells. We demonstrate here that mammalian cells infected with LCMV *in vitro* release particles containing LCMV GP1 in a format that is accessible to polyclonal B cells. These particles possess a mass greater than 200kDa and are thus likely membrane bound vesicles and not free proteins.

## Viral replication is dispensable for polyclonal B cell acquisition of LCMV antigen

To eliminate the possibility that B cells are acquiring LCMV antigen intracellularly by being productively infected, we performed our TBAP using non-replicative, UV-killed (UVK) LCMV. If instead of acquiring antigens from the extracellular milieu, B cells gain access to LCMV antigen through intracellular viral replication then using UVK LCMV would render B cells incapable of presenting LCMV antigens to T-cells. LCMV infected cell supernatants were subjected to UVC irradiation until no focus forming units could be detected by focus-forming assay. B cells incubated overnight with UVK LCMV and subsequently co-cultured with SMARTA-I T cells had significantly increased levels of CD86 expression compared to negative controls (Fig 2B). All other metrics of antigen presentation; T-cell proliferation and cytokine production, were also significantly increased compared to controls (Fig 2A-E). This result demonstrates that polyclonal B cells can acquire LCMV antigens from their extracellular environment independently of viral replication.

## LCMV GP1 is present on infected cell-derived particles smaller than 50 nm

We wanted to extend the size range of our TBAP beyond the kDa range to include any particle smaller than a complete LCM virion. LCMV is often described as being 110–130nm in diameter however studies have reported virions as large as 300nm or as small as 50nm in diameter [18]. To be as conservative as possible we passed our LCMV-infected cell supernatant through a 20nm (0.02μm) syringe filter. To our surprise, we detected viral replication in the 20nm filtrate using our focus-forming assay. We labeled the 20nm filtrate with the anti-GP1-specific WEN3 mAb and gold bead secondary antibody for further investigation using transmission electron microscopy (TEM) (Fig 3A). We observed various sizes of particles containing LCMV GP1 on their surface (Fig 3B) some of which are large enough to be intact virions. These results made it clear that the 20nm filter was not effective at separating virus from non-infectious sources of antigens but also proved that particles containing LCMV-GP1 are present at sizes less than 50nm. Interestingly particles from uninfected cells (Mock) showed similar morphology and size distribution to antigen bearing particles released from infected cells, suggesting not all the antigen bearing particles are likely to be virus but could also include antigen-bearing exosomes.

## Antigen is available in both high and low viral titer fractions

In another attempt to separate infectious from non-infectious sources of viral antigen, we used a glucose density gradient centrifugation approach. Specifically, we sought to test whether B cells acquired LCMV antigen from non-infectious sources or from intact, infectious LCM virions. We performed a rate-zonal centrifugation of LCMV-infected cell culture supernatant using a 0–20% (w/v) sucrose step gradient. Discreet fractions of glucose-antigen mixtures at varying densities were collected and tested for infectivity in focus-forming assays. Although, as others have shown [13,19–23], it was not possible to obtain a fraction completely devoid of infectious activity, we determined that the majority of infectious LCMV material was contained in fractions with a density greater than 15% glucose (w/v). Viral titers in the 15–20% fractions varied from 1x10^4 to 3x10^4 ffu/ml. Viral titers in glucose fractions <15% varied from undetectable levels to 1x10^3 pfu/ml. For our TBAP we used three fractions; the pooled 15 and 20% fraction labelled "high-titer" or "infectious," the low-titer 10% fraction and the low-titer 5% fraction. Each fraction was buffer changed to RPMI complete with a 100kDa MWCO filter to remove the glucose. We hypothesized that if B cells were unable to access Ag from infectious viral particles then we would expect there to be less antigen presentation when given the high titer fractions.

Notably, all fractions, regardless of viral titer and density contained sufficient antigen to allow significant levels of antigen presentation compared with the negative control (Fig 4A–E). This includes measurements of MHC-II expression levels in B cells which were significantly elevated in both high and low titer fractions with respect to mock (S4 Fig). Expectedly, we observed that the process of centrifugation and separation of infected cell supernatants through density gradients reduced antigen presentation metrics compared to the uncentrifuged supernatant as some antigen is lost in the separation and collection steps as well as the buffer change (Figs 4A–E and S4). Collectively, our antigen presentation metrics did

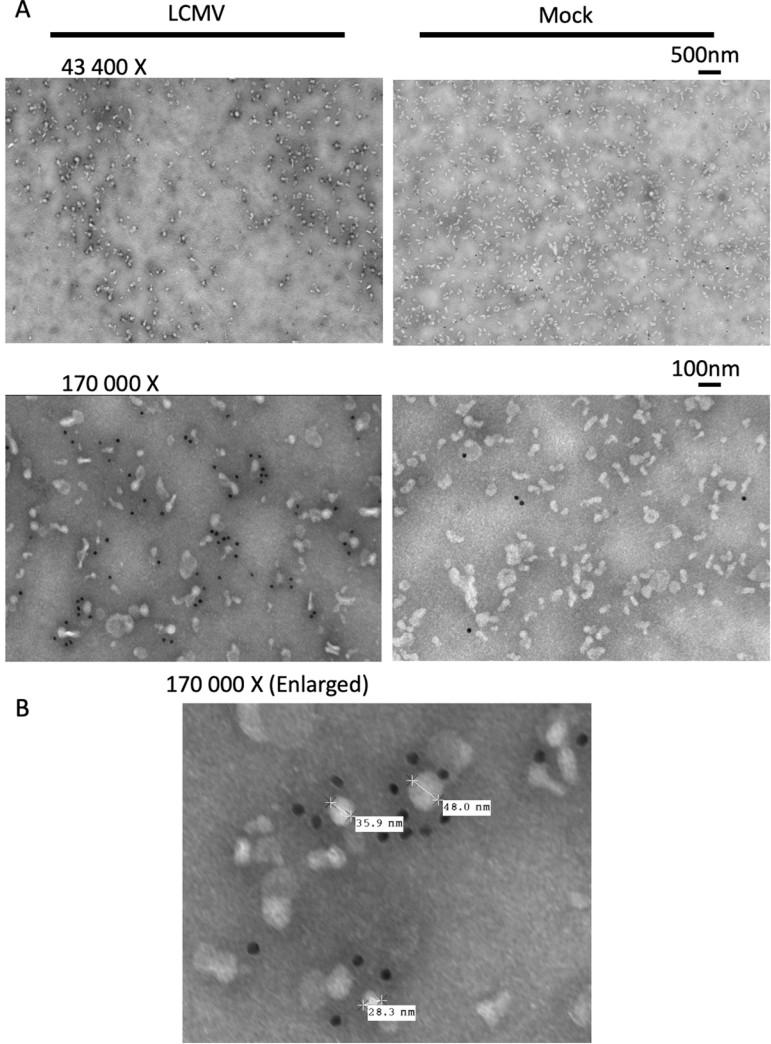

**Fig 3. LCMV GP1 is present on infected cell-derived particles smaller than 50nm.** (A) Transmission electron microscopy (TEM) image of particles derived from LCMV-infected (LCMV, left) or uninfected (Mock, right) cell supernatant labeled with WEN3 primary antibodies and 10nm gold bead-conjugated secondary antibodies (black dots), at 43 400X magnification (top) and 170 000X magnification (bottom). (B) TEM image of LCMV-infected cell supernatant showing particle lengths measured in nm at 170 000X magnification.

not point to any one fraction having more accessible antigen than another. Our results suggest that while some Ag may be acquired from low density/non-infectious sources, it remains a possibility that B cells could acquire Ag from infectious virus.

## Polyclonal B cells proliferate after presenting antigens acquired *in vitro* in LCMV-infected mice

We wanted to test whether the antigens B cells acquired from infected cell cultures *in vitro* could be presented to CD4+T cells in an infected mouse *in vivo*. T cells from infected mice recognize a wider array of antigenic epitopes than SMARTA-I T cells and, the interaction between B cells and T cells would be influenced by the cytokine milieu, lymphoid structure and many other *in vivo* conditions absent in our model. To do this, we adapted our TBAP assay to an *in vivo* adoptive transfer assay. CFSE labelled polyclonal B cells from uninfected CD45.2 donor mice were cultured overnight in antigens before

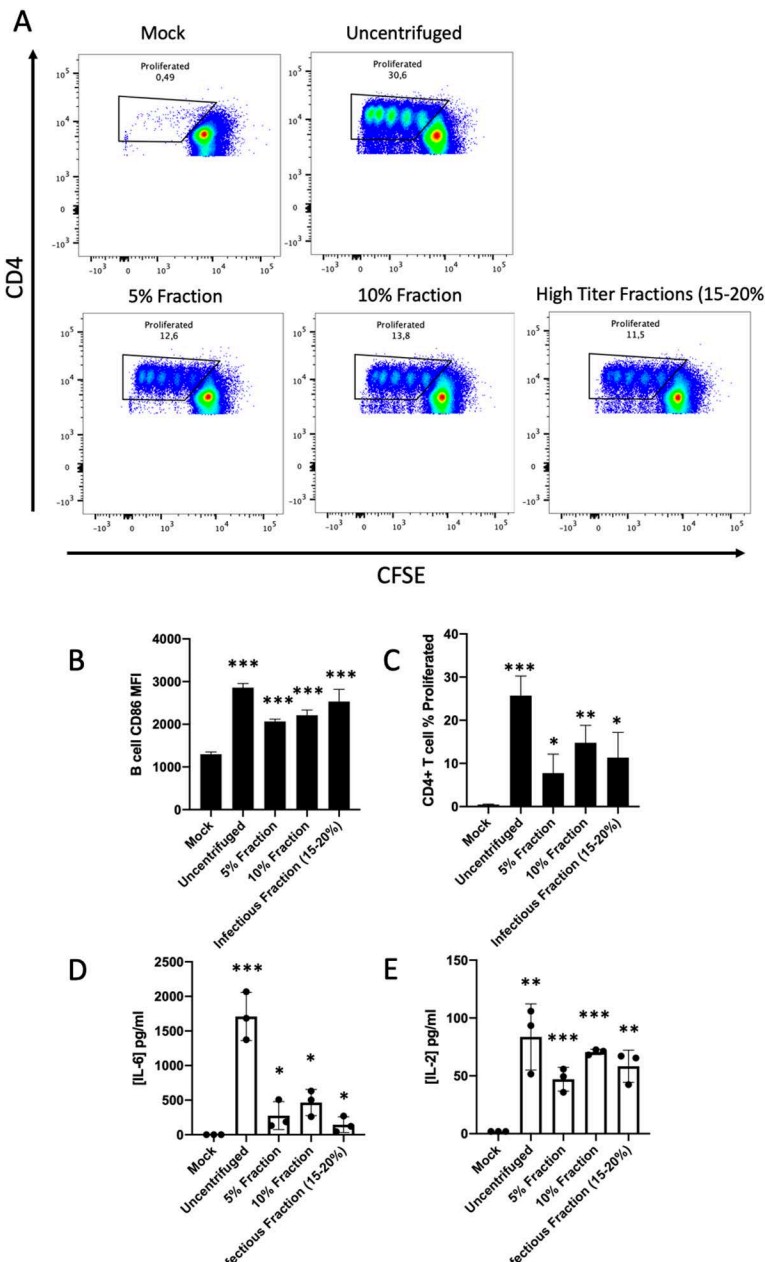

**Fig 4. Antigen is available in both high and low viral titer fractions.** (A) FACS plots of CFSE labeled SMARTA-I CD4+T cells after 72 hours of co-culture with antigen loaded wild-type B cells, each plot shows data from one mouse, and each assay was independently repeated a minimum of three times ($n = 3$). (B) CD86 expression on B cells measured by mean fluorescent intensity. (C) Percentage of SMARTA-I CD4+T cell proliferation measured by gating on CFSE low cells (example gate seen in panel A). (D) IL-6 concentration in pg/ml detected in 72 hour co-culture medium measured by ELISA. (E) IL-2 concentration in pg/ml detected in 72 hour co-culture medium measured by ELISA. In all bar graphs values are averages of three experiments ± SEM, * denotes significance ($P < 0.05$) with respect to Mock in an unpaired student's T-test.

being injected intravenously into CD45.1 recipient mice, which had been infected with LCMV 8 days prior. One week after adoptive transfer, total splenic B cells from recipient mice were collected and CD45.2+B cells were analysed for proliferation via CFSE loss. We observed B cell proliferation *in vivo* in all the same conditions where we had previously observed

SMARTA-I CD4+T cell proliferation *in vitro* (Fig 5). Importantly, at the timepoint B cells from infected mice were collected, donor B cells loaded with mock supernatant *in vitro* did not proliferate. The latter indicates that donor B cells are presenting the antigens with which they were loaded *in vitro*, and not newly captured antigen from the *in vivo* environment. These results suggest that the polyclonal B cells from our *in vitro* model can present antigen, receive T cell help, and proliferate in the context of LCMV infection *in vivo*.

### Inhibition of exosomes or DI particle release does not significantly reduce B cell antigen presentation

We next decided to change our approach and instead of attempting to remove infectious virus from our samples, we decided to target potential sources of non-infectious antigen. DI particles are non-infectious, produced during the course of normal viral replication, and can constitute a considerable proportion of viral antigen released from cells [13]. To test if these particles constitute an important source of non-infectious antigen available to B cells we performed our TBAP assay using supernatants from cells infected with wild-type (WT) LCMV (LCMV-PPXY) alongside cells infected with the recombinant LCMV-AAAA described by Ziegler and colleagues [13]. The LCMV-AAAA recombinant virus no longer encodes LCMV's lone PPXY late domain (PPXY→AAAA) on its Z protein resulting in undetectable levels of DI particle formation [13,24]. Because the supernatant from cells infected with WT LCMV had a 50-fold higher viral titer compared with that from the LCMV-AAAA mutant (S2 Fig) and because whole LCMV virus remains a potential source of antigen it was necessary for us to dilute the LCMV-PPXY (WT) by 1 in 50 to control for infectious viral content and allow a meaningful comparison of total antigen content. We did not observe a significant reduction in antigen presentation when B cells were co-cultured in supernatant from LCMV-AAAA infected cells compared with supernatant from unmutated LCMV-PPXY

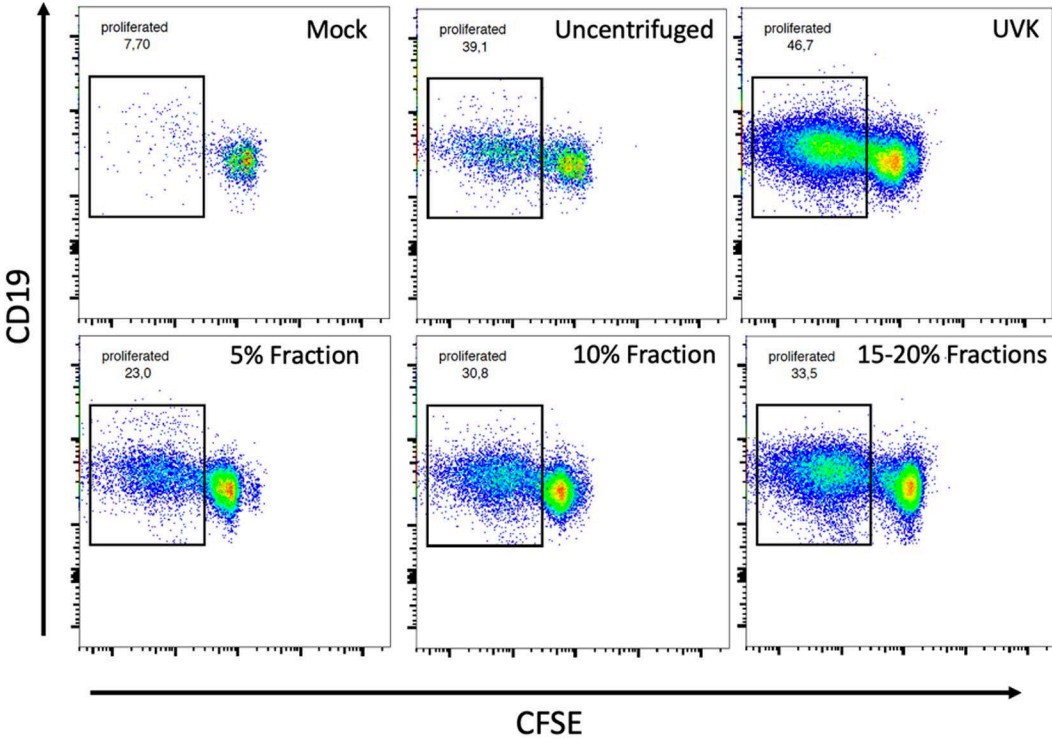

**Fig 5. Polyclonal B cells proliferate after presenting antigens acquired *in vitro* in LCMV-infected mice.** FACS plots of antigen loaded, CFSE labeled CD45.2 B cells one week after being adoptively transferred into D8 LCMV-infected CD45.1 mice, each plot shows data from one mouse, the assay was independently repeated twice.

(Fig 6A, 6C-D and 6G-H). However, we did observe a subtle difference that very nearly approached statistical significance in the extent of proliferation (Pvalue = 0.052), as measured by proliferation index as opposed to % proliferation, between AAAA and PPXY LCMV (S5 Fig). From this, we can conclude that while DI particles are not minimally required for B cell access to LCMV antigen they could possibly act as a redundant source of antigen.

Another potential source of non-infectious antigen are exosomes released from infected cells containing viral protein but lacking the viral genome. To determine whether exosomes released from infected cells could act as an antigen source for polyclonal B cells, we used the exosome inhibitor GW4869 on BHK cells productively infected with LCMV. Similar to results obtained by others [25] we found that 2uM of GW4869 did not negatively impact BHK cell survival nor did it result in drastically different viral titers than infected DMSO control BHK cells (ARM with GW4869: 2.2x10^6 ffu/ml versus ARM with DMSO: 3x10^6 PFU/ml). We then performed a TBAP assay with supernatants from infected cells treated with or without GW4869. While we did observe a tendency towards reduced metrics of antigen presentation in conditions with GW4869 compared with the DMSO control the difference was not statistically significant (Fig 6B, 6E-F and 6I-J). While our results do not exclude DI particles or exosomes as sources of antigen for B cells, they indicate that neither source is alone essential and that B cells are able to acquire sufficient levels of antigen from other sources.

### Anti-LCMV GP antibodies inhibit antigen presentation by polyclonal B cells *in vitro*

Because our results thus far have not enabled us to exclude whole virus as a source of antigen accessible to polyclonal B cells, we began to ask how it might be possible for B cells to internalize intact virus. Either sufficient <50nm virus is present to allow B cells to acquire infectious virions through pinocytosis alone, or B cells possess a receptor enabling receptor-mediated entry of virions of all sizes. LCMV is an enveloped virus and therefore may carry over many membrane-bound receptors when budding from its host cell that could interact with B cells [26,27]. Because the GP-1 trimer is the primary ligand associated with LCMV entry into the permissive cells [28] we reasoned this was the best candidate to study first. We therefore sought to test if by blocking GP-1 with a neutralizing antibody we would observe diminished antigen presentation. For this, we used the IgG1 monoclonal KL25 antibody, a neutralizing antibody that recognizes an epitope found apically on the GP1 protein of the WE LCMV strain [29]. We used the same TBAP as previously described except we used a recombinant LCMV virus; a Cl13 strain that expresses the GP of WE [30,31]. Our choice to use a WE glycoprotein-expressing LCMV Cl13 instead of the LCMV WE strain itself was a pragmatic one. LCMV WE is a biosafety level (BSL) 3 strain of LCMV while the Cl13 variant of ARM is BSL2 allowing us to manipulate the latter strain safely in our BSL2 facility. ARM and rCl13 WE-GP were both produced on the same cell lines and we did not observe any biological differences between these two strains within the parameters of our study except for the ability of the latter to bind KL25. We incubated the recombinant virus with the KL25 antibody for 40min at 4°C before combining the virus/antibody mix with polyclonal B cells overnight. To avoid B cell interactions with the Fc region of the KL25 antibody we exposed B cells to Fc receptor-blocking antibodies before and during overnight culture with the antigen/antibody mix and with the "virus only" control. To control for the effects of KL25 bivalency which is known to lead to virus amalgamation and immunocomplex formation [31] we also tested F(ab) generated from the digestion of KL25 IgG1 with papain. We observed a significant reduction in both the proportion of proliferating T cells and in the expression of B cell CD86 as a result of both full KL25 IgG and KL25 F(ab) addition to the recombinant LCMV virus compared to conditions with the recombinant virus alone (Fig 7A-D). These results indicate that polyclonal B cells are limited in their ability to acquire antigen if GP1 is bound by the KL25 neutralizing antibody indicating the involvement of a GP1-specific receptor on polyclonal B cells responsible for antigen acquisition and presentation to CD4 + T cells.

### Discussion

Hypergammaglobulinemia during LCMV infection is the result of antigen presentation between polyclonal B cells and LCMV-specific cognate CD4+ T cells [1]. In the present study we developed an *in vitro* model to test several hypotheses

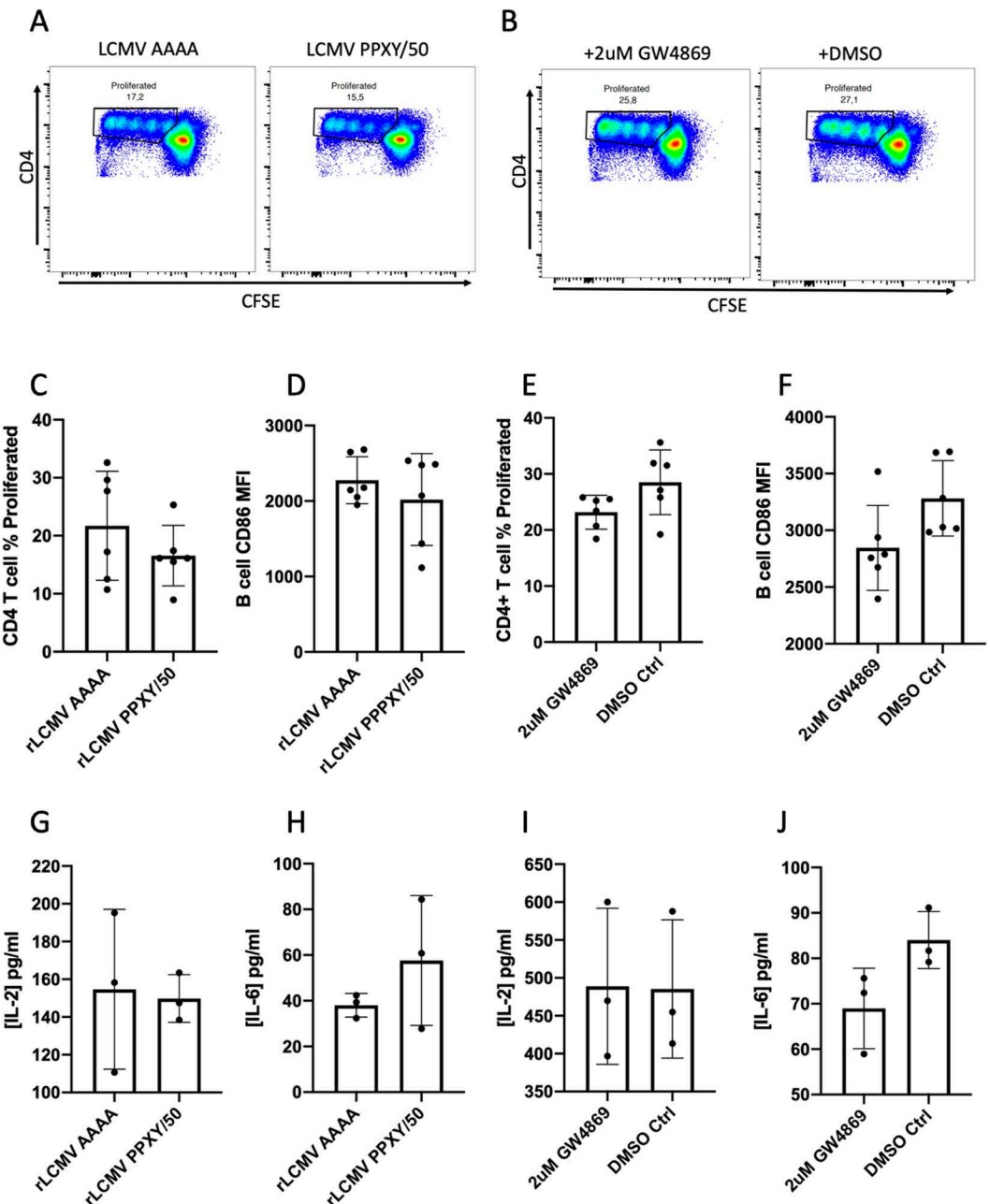

**Fig 6. Inhibition of exosomes or DI particle release does not reduce B cell antigen presentation.** (A) FACS plots of CFSE labeled SMARTA-I CD4+T cells after 72 hours of co-culture with wild-type B cells loaded with antigen from either WT LCMV PPXY or the DI particle null virus called LCMV AAAA. (B) FACS plots of CFSE labeled SMARTA-I CD4+T cells after 72 hours of co-culture with wild-type B cells loaded with antigen from LCMV infected cells treated with either DMSO or the exosome inhibitor GW4869. Each plot shows data from one mouse, and each assay was independently repeated a minimum of three times (*n* = 3). (C, E) Percentage of SMARTA-I CD4+T cell proliferation measured by gating on CFSE low cells (example gates seen in panels A and B). (D, F) CD86 expression on B cells measured by mean fluorescent intensity. (G, I) IL-2 concentration in pg/ml detected in 72 hour co-culture medium measured by ELISA. (H, J) IL-6 concentration in pg/ml detected in 72 hour co-culture medium measured by ELISA. In all bar graphs values are averages of three experiments±SEM, * denotes significance (P<0.05) with respect to Wilt-type LCMV (PPXY/50) or DMSO control respectively in an unpaired student's T-test.

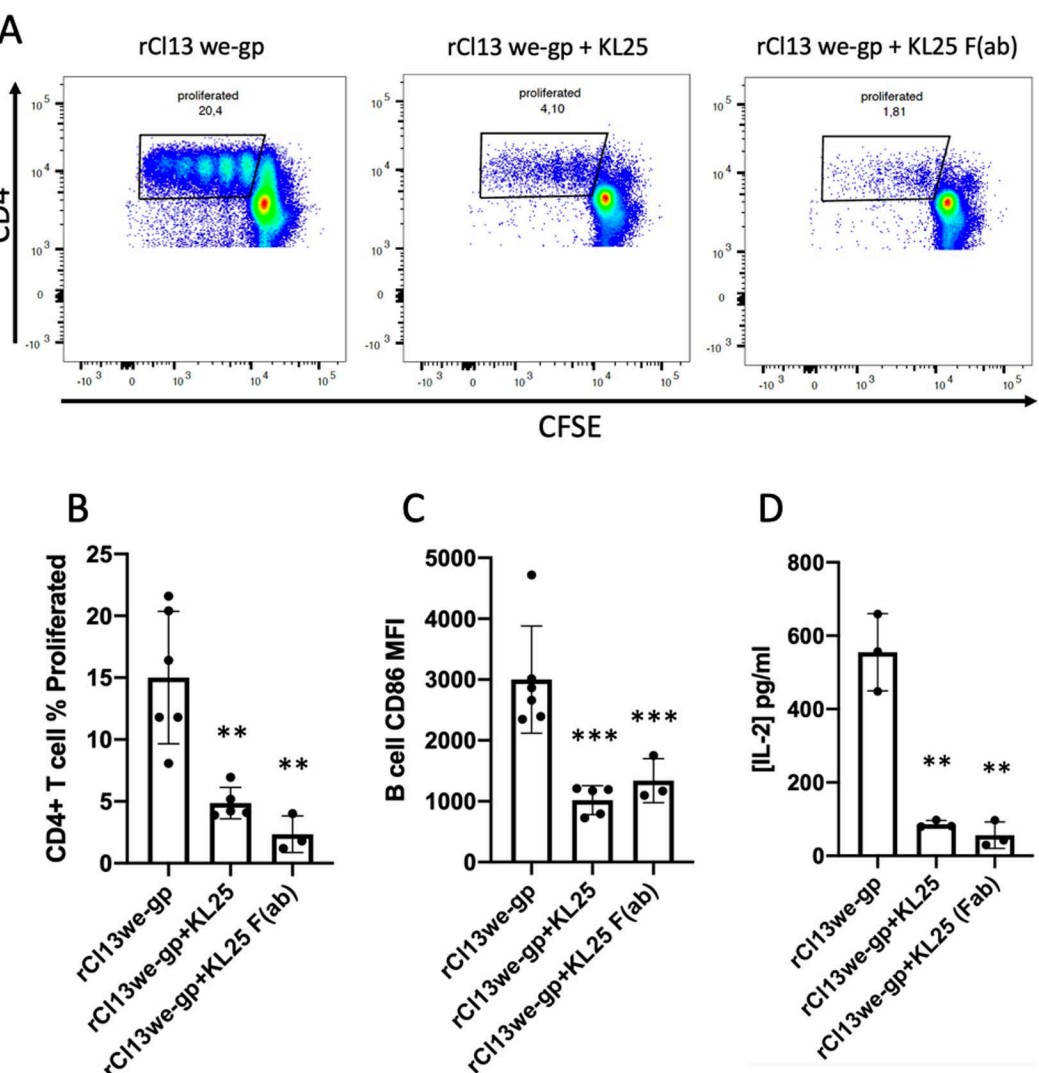

**Fig 7. Anti-LCMV GP antibodies inhibit antigen presentation by polyclonal B cells in vitro.** (A) FACS plots of CFSE labeled SMARTA-I CD4+T cells after 72 hours of co-culture with antigen loaded wild-type B cells, each plot shows data from one mouse, and each assay was independently repeated a minimum of three times ($n=3$). (B) Percentage of SMARTA-I CD4+T cell proliferation measured by gating on CFSE low cells (example gate seen in panel A). (C) CD86 expression on B cells measured by mean fluorescent intensity. (D) IL-2 concentration in pg/ml detected in 72 hour co-culture medium measured by ELISA. In all bar graphs values are averages of three experiments ± SEM, * denotes significance ($P < 0.05$) with respect to rCl13WE-GP in an unpaired student's T-test.

concerning the manner in which polyclonal B cells acquire LCMV antigens as well as the nature of the antigens acquired. We use antigen presentation by polyclonal B cells to LCMV specific SMARTA-I T cells as an indicator of antigen acquisition. Consequently, we focused our analysis on well-defined changes in B cells and T cells when antigen presentation occurs. We did not see changes in the metrics we measured when B cells were given LCMV antigen and exposed to non-cognate CD4 T cells from uninfected wild-type B6 mice (S1A Fig). The latter suggests our model captures *bonafide* antigen presentation rather than a direct B cell response to LCMV antigen that might result an antigen-unspecific bystander response in T cells. We found that antigens released by cells infected with LCMV *in vitro* are available for uptake by polyclonal B cells demonstrating that NK/CD8+ T cell mediated destruction of infected cells is not required for

Ag release (Fig 2A-E). Similarly, we found that antigen presentation alone was sufficient for robust co-activation of poly-clonal B cells and antigen-specific CD4 + T cells, demonstrating that the cytokine milieu present during *in vivo* LCMV infection is not essential for this co-activation to occur (Fig 2A-E). Using adoptive transfer, we show that B cells that acquire antigens *in vitro* are able to present these same antigens *in vivo* and subsequently proliferate (Fig 5). Importantly, B cells adoptively transferred into LCMV infected mice did not proliferate unless they were given LCMV antigen prior suggesting they are likely responding to cognate T cell help rather than the cytokine milieu or some other signal *in vivo* (Fig 5). Our results confirm findings in previous studies that demonstrate how polyclonal B cells which acquire LCMV antigens in a BCR-independent manner *in vitro* will proliferate [6]. The same study shows that polyclonal B cells loaded with the LCMV MHC-II immunodominant epitope GP61 produce polyclonal antibodies when adoptively transferred to LCMV infected mice, demonstrating that HGG is a result of their ability to acquire LCMV antigen BCR-independently [6]. The novelty of our result lies in that we used LCMV antigen derived from infected cells instead of recombinant antigenic peptides thus more realistically modelling the type of antigen and uptake B cells encounter *in vivo.* These results taken together allow us to conclude that infected cells *in vivo* release antigens in a manner and form that is well represented by our *in vitro* model. Our study also suggests that while non-infectious antigen sources such as antigen laden exosomes and DIPs could potentially play a redundant role (Figs 4 and 6) LCMV virions are sufficient sources of viral Ag for polyclonal B cell-based activation of CD4 + T cells (Fig 4). Finally we provide evidence suggesting that these virions likely enter B cells via receptor-mediated endocytosis (Fig 7).

We found no evidence of protein sized antigen release smaller than 200kDa using immunodetection (Fig 1A-C) or antigen presentation assays (Fig 2A-E). This allows us to exclude GP shedding as a mechanism of antigen release by infected cells *in vitro*. However, if GP-shedding does occur *in vivo* our results show that polyclonal B cells can readily access sufficient quantities of LCMV antigens for robust co-activation with cognate helper T cells even in the absence of shedding. There are other mechanisms by which infected cells may release antigen-laden particles other than infectious virions. LCMV infection is known to result in the significant release of defective interfering particles (DI particles/ DIPs) [32,33] as well as exosomes containing viral antigens [34,35]. With TE microscopy we identified LCMV GP-bearing particles at various sizes including <50nm (a size accessible to B cells via pinocytosis). Exosomes and DIPs are morpho-logically indistinguishable from infectious virus and our data suggests that these classes of particles are not minimally required to drive B-cell mediated activation of LCMV-specific CD4 + T cells (Figs 6 and S2). Despite this, we still believe that these non-infectious antigen-bearing particles are available to B cells (Fig 4) and that they act as redundant antigen sources along with whole virus. These results changed our presumption regarding the nature of the antigen available to B cells. We now had evidence that suggested infectious virus particles may be an important source of antigen for polyclonal B cells.

Most LCMV virus is roughly 100nm in diameter [18] making it a large particle for pinocytosis by B cells. Though not much is known regarding the size limits of receptor-independent endocytosis in B cells, studies on their ability to internal-ize nanoplastics give us a rough estimate of 50nm in diameter as a permissible size for pinocytosis [7]. Either polyclonal B cells are able to take up sufficient quantities of LCMV on the smaller end of their size range or these B cells may be using a receptor mediated form of endocytosis. We used the neutralizing anti-LCMV GP antibody KL25 to show that virus opsonization reduces polyclonal B cell access to LCMV antigen (Fig 7A-D). Using KL25 IgG digested to monovalent F(ab) (Fig 7A-D) we show that the ability of KL25 to reduce B cell access to antigen cannot solely be the result of virus amalgamation into immune complexes too large for pinocytosis, but rather the result of the specific blocking of GP on the virus. We suggest that our results indicate the presence of a receptor on B cells able to interact with the LCMV GP trimer enabling polyclonal B cells to internalize whole virus.

A recent study using a non-cleavable version of the LCMV GPC ectodomain combined with an isolated GP1 domain as a probe did not detect any GP1-binding B cells in either uninfected mice or LCMV-immune mice 28 days after infection suggesting that GP1 binding B cells are very rare [36]. Conversely our study shows that polyclonal B cells from naïve mice

that can acquire LCMV antigen are abundant and that antigen acquisition depends largely on the accessibility of GP1. This discrepancy might be explained by the nature of the probe used. It is well described that LASV GP1 monomers do not bind alpha-dystroglycan (α-DG), [37–40]. Binding to the LASV canonical receptor depends on the quaternary structure formed by the trimerization of GP-1 subunits [41]. The LCMV GP1 trimer shows a strikingly similar quaternary structure for matriglycan (the polysaccharide present on α-DG recognized by GP-1) recognition [42]. Therefore, while an LCMV GP1-GPC probe in monomeric form will bind Ig/BCR as expected it is unlikely to bind receptors that depend on quaternary structural conformation. It may also be possible LCMV endocytosis may depend on more than one receptor. Receptors for host proteins present on the viral envelope may also play a role in allowing virions to stick to the B cell surface long enough for a GP1-dependent receptor to be engaged thus allowing endocytosis. For example, ERGIC-53 is a host protein that is incorporated into arenavirus particles and facilitates their attachment to subsequent host cells thereby increasing their infectivity [43]. Similarly, endocytosis may depend on receptor clustering on B cells, or GP1 trimers may have weak avidity for their ligand on B cells, in both cases the polyvalency of multiple GP-1 trimers displayed on the virion membrane could be needed to allow antigen entry.

Alpha-dystroglycan, is the canonical receptor for LCMV GP1 [44], and is expressed at low levels on B cells (S3 Fig) however α-DG isolated from lymphocytes show poor binding to LCMV [4]. Alternative receptor-mediated modes of entry, such as by FcR or phosphatidylserine (PS) receptors, have also been proposed [26,27]. Indeed, although the overall virulence of LCMV correlates with the strain's affinity to its receptor; alpha-dystroglycan (α-DG) [4,45–47], the presence of α-DG does not always correlate with cell infectibility [48,49]. Differences in the *in vivo* and *in vitro* infectability of cells by LCMV Cl13 have also led to speculation on the existence of co-receptors or alternative modes of entry [26]. Similarly, the use of receptors other than α-DG has been described in the related old-world arenavirus Lassa (LASV) [50,51]. Some alternative receptors described in LASV infection have similarly been tested as potential receptors for various strains of LCMV [52]. Another important consideration is how glycosylation of LCMV GP affects viral tropism and binding to host cells [53,54]. It could be that polyclonal B cells possess a receptor that binds LCMV GP1 indirectly via the various N-linked glycans on the GP1 trimer. Many of the aforementioned studies focused on dendritic cells (DCs) and macrophages as these cells are known to be susceptible to infection with LCMV Cl13, however, whether these alternative receptors might be used by polyclonal B cells to acquire antigens for presentation to cognate helper T cells remains untested. Therefore, the identity of the receptor we propose in our study requires further exploration.

Many studies have shown that B cells are refractory to infection by LCMV [1,4,5] nevertheless we included the infection hypothesis in our assay. We show that polyclonal B cells can acquire antigen from non-replicative UVC-irradiated LCMV (Fig 2A-E) demonstrating that antigen acquisition does not rely on intracellular infection of polyclonal B cells. If the receptor we propose exists it begs the question as to why LCMV does not replicate upon entry to B cells. LCMV membrane fusion with the endosome and subsequent release of viral RNA is pH-dependent. It is possible that LCMV does not find an intracellular environment suitable for replication; if the endosome enclosing the virus is not sufficiently acidic for the release of its RNA before being fused with the late endosomal-lysosomal antigen-processing compartment the result would simply be the proteolysis of the viral antigens and MHC-II loading without intracellular replication. Identifying the receptor on B cells that interacts with LCMV-GP1 would allow a more detailed study on the fate of internalized virus.

In conclusion, we found evidence supporting the possibility of pinocytotic antigen acquisition however we show here for the first time that receptor-dependent antigen acquisition by polyclonal B cells plays a significant role. Blocking LCMV-GP1 using a neutralizing antibody resulted in a significant reduction of B cell antigen presentation leading us to believe that receptor mediated endocytosis is the principal means to antigen acquisition. Currently there is no feasible means of blocking all forms of pinocytosis without also disrupting receptor mediated entry. Both forms of endocytosis use various and often overlapping mechanisms for internalization. Furthermore, inhibitors that block receptor mediated entry of LCMV antigen to B cells would necessarily also risk preventing LCMV from entering permissive cells thereby making it impossible to differentiate the effects of blocking B cell antigen access from the effects of reducing viral titres on HGG. Along

the same reasoning it is not possible to directly test the antigen access blocking effects of KL25 on B cells *in-vivo* as we have done here *in vitro* as KL25 is known to reduce viral load [31]. For these reasons our efforts in this study focused on exclusion by antigen size, density and release mechanism and our *in vitro* model to draw conclusions regarding the pinocytosis hypothesis. Further studies to identify the LCMV-GP1 receptor on B cells as well as measure LCMV binding to B cells are still necessary to confirm our hypothesis. Once the identity of the receptor on B cells that interacts with GP1 is elucidated it will become possible to block it specifically *in vivo* and determine its effects on HGG levels. Although the general link between HGG and polyclonal B cell BCR-independent acquisition of LCMV antigen has already been made by others [1,6], our study is the first to specifically test the mechanism of LCMV antigen acquisition by polyclonal B cells. The findings in our study are the first to suggest that whole virus is accessible to polyclonal B cells and that this access is dependent on binding with viral GP1.

## Materials and methods

### Ethics statement

This study was conducted in accordance with the guidelines of the Canadian Council on Animal Care. All animal experiments were reviewed and approved by the institutional animal care and use Committee; Comité Institutionnel de Protection des Animaux (CIPA) of the Institut National de la Recherche Scientifique – Centre Armand Frappier Santé et Biotechnologie (CIPA Protocol no. 2209-05).

### Animals, viruses, proteins and cell lines used

SMARTA-1, transgenic mice containing a CD4+ T cell repertoire that is specific for the LCMV epitope GP61–80 [55], and non-carrier littermates, were obtained from Jackson Laboratories, (JAX stock #030450). For adoptive transfer assays, C57BL/6J mice carrying the CD45.2 allele (JAX stock #000664) and B6 CD45.1 mice (JAX stock #002014) were obtained from Jackson Laboratories. rLCMV Cl13 WE-GP; a virus with LCMV Clone 13 backbone and glycoprotein from the WE strain, was a generous gift from Dr. Daniel D. Pinschewer at the University of Basel, Switzerland, via the European Virus Archive global (EVAg) consortium (Ref-Sku:007V-03891). The virus was produced on BHK21 cells (ATCC CCL-10). rLCMV-AAAA; as described in [24] is a variant of LCMV Armstrong 53b where the late domain of the Z protein was mutated from PPXY to AAAA and consequently does not produce detectable levels of defective interfering particles. The mutant virus and the rescue control; rLCMV-PPXY, were both produced in Vero cells. LCMV Armstrong 53b (originally obtained from Dr. Rolf M. Zinkernagel) was produced on BHK21 (ATCC CCL-10), MC57G (ATCC CRL-2296) or L929 cell lines (ATCC CCL-1) originally obtained from Dr. Pierre J. Talbot. For assays testing the effects of exosomes, LCMV Armstrong infected BHK cells supplemented with exosome depleted FBS (Gibco) were cultured in either in the presence of 2uM of GW4869 (Selleckchem S7609) from a stock solution 1mg/ml in DMSO or with an equal volume of DMSO only. Supernatants from uninfected BHK21 or L929 cells were collected for mock conditions. MC57G was used to determine viral titres in plaque assays using the protocol described by Welsh and Seedhom [56]. Soluble recombinant LCMV WE glycoprotein (Creative Biolabs Cat# GPX04-275J) contains the immunodominant GP61–80 amino acid region as well as the region containing Asn119 responsible for KL25 binding. It was produced using a Baculovirus expression host and purified using immobilized-metal affinity chromatography.

### Treatment of infected cell supernatants

In order to concentrate virus, change buffers or separate viral proteins by size, supernatant from LCMV-infected cells were passed through either Amicon-Ultra 100kDa, Microcon-10 10KDa, ADVANTEC 200kDa, Whatman Anotop 0.02μm or MillexGP 0.22μm filter units according to the respective manufacturer's instructions. Filtrates were collected and retentates, where possible, were resuspended by gently washing the filter membrane with RPMI.

For conditions using UV-killed, non-replicative, LCMV; the infected cell supernatant was irradiated with UVC for 2 minutes and the absence of infectivity was verified by viral plaque assay on MC57G cells.

Virus and viral proteins in the supernatant of LCMV-infected cells were separated by density with a glucose step gradient of 0–20% glucose (w/v). Briefly, a solution of 20% (w/v) glucose in 0.01M Tris 0.1M NaCl 0.001M EDTA (disodium) buffer was prepared and sterilized through filtration with a MillexGP 0.22μm filter unit. 5%, 10% and 15% (w/v) sucrose solutions were then prepared by adding buffer to 20% glucose solution at a 3:1, 1:1 and 1:3 ratio respectively. Trypan blue was added to alternating glucose solutions to enable visual distinction. Equal volumes of glucose solutions were carefully added to ultracentrifuge tubes using a syringe from lowest concentration to highest, starting with buffer only (0%). For a single 15% glucose gradient, a 3:1 ratio of 20% (w/v) glucose was mixed with buffer. The infected cell supernatant was then added directly to the top layer prior to ultracentrifugation at 20 000rpm for 75 minutes in an Optima L-100 XP ultracentrifuge with SW41 swinging bucket rotor (Beckman Coulter). After centrifugation each fraction was collected with a syringe ensuring not to disturb the interphase between fractions. Each glucose fraction was subsequently passed through a Amicon-Ultra 100kDa filter unit to buffer change antigens from glucose to RPMI complete and then preserved at -80°C for subsequent use in assays.

## T-cell B-cell Antigen Presentation (TBAP) assay

Spleens from B6 (SMARTA-1 non-carrier) mice were collected and splenocytes were passed through a 100μm cell strainer (Corning product #431752). Polyclonal B cells were isolated by negative magnetic bead selection according to the manufacturer's specifications (Stemcell Technologies, Catalog #19854). A minimum of 1x10^6 B cells are incubated overnight in RPMI complete with either mock (supernatant from uninfected L929 or BHK cells) or 300μg/ml of LCMV GP61–80 peptide (New England Peptide) for negative and positive controls respectively. For LCMV conditions, B cells were incubated overnight in 500μl of supernatant from LCMV infected cells that had been buffer exchanged to RPMI complete as described above. For filter validation experiments, 100μg/ml of soluble recombinant LCMV-WE glycoprotein in complete RPMI was filtered using Amicon Ultra 100kDa filters. The retentate was resuspended in 500μl of complete RPMI and diluted 1 in 2, alongside the filtrate. A total of 500μl of each preparation was used for overnight incubation with B cells.

After overnight incubation with LCMV antigen, B cells are washed in warm RPMI complete and co-cultured with a minimum of 2x10^6 CFSE labelled LCMV specific CD4 T-cells isolated from SMARTA-I positive transgenic mice. CD4+T cells were similarly isolated using negative magnetic bead selection according to the manufacturer's specifications (Stemcell Technologies, Catalog #19852). Following 72hrs of co-culture, the cells were centrifuged, the culture media was collected for cytokine detection by ELISA and the cells were labelled with fluorescent antibodies for analysis with flow cytometry.

## Adoptive transfer antigen presentation assay

On D0 CD45.1 B6 mice were intraperitonially injected with 2x10^6 PFU of LCMV Armstrong. The day before the adoptive transfer, splenic B cells (~1x10^7 cells) from a CD45.2 B6 donor mouse were isolated using negative selection magnetic beads, labelled with CFSE, and then incubated in their respective antigen solution overnight (as described in the TBAP assay above). Importantly, FBS was removed from the antigen solutions via buffer exchange to FBS-free RPMI complete using the Amicon 100kda filter. B cells were and then washed in PBS and injected intravenously into D8 LCMV Armstrong-infected mice. Seven days after adoptive B cell transfer, spleens from recipient mice were collected and B cells isolated as described above. CD45.2+B cell proliferation was then measured via CFSE loss.

## KL25, KL25 Fab antibody interference TBAP

KL25 monoclonal antibody [28] was purified from KL25 hybridoma cell culture supernatants using HiTrap Protein G HP (Cytiva product #29048581) on the Cytiva Äkta Prime system. KL25 Fab was produced from purified KL25 IgG using the

Pierce Fab Preparation Kit (Catalog #44985) according to the manufacturer's instructions. Complete separation of Fab from Fc fragments was confirmed by non-reducing western blot following the manufacturer's instructions. The rCl13 WE-GP virus and KL25 antibody were tested together at various concentrations to determine the best ratio allowing the effects of the antibody to be seen without requiring prohibitively large volumes of antibody. We found that a 1 in 50 dilution of our virus (titered at 5.7x10^6 ffu/ml) allowed for sufficient B cell antigen presentation to observe measurable differences when coupled with KL25 antibody. In TBAP conditions using KL25 or KL25 Fab; 10µl of supernatant from rCl13 WE-GP LCMV infected cells were mixed directly in 50µl of purified KL25 or KL25 Fab (1mg/ml) for 40 minutes at RT. The antigen-antibody mixture was then brought to 500ul in RPMI complete and presented to B cells overnight as described in "TBAP Assay" above.

## Immunodetection assays

Dot-blots; Supernatant from LCMV cells was collected at the various indicated time points. A 15µl drop of each supernatant was blotted onto pre-rinsed PVDF or Nitrocellulose membranes and either allowed to absorb by gravity overnight at 4°C or by aspiration using the Bio-Rad Bio-Dot apparatus. Membranes were then blocked and probed with hybridoma supernatant containing WEN3 (anti-LCMV GP) or VL4 (anti-LCMV NP) antibodies [29]. Blots were revealed using the Bio-Rad ChemiDoc imaging system. ELISAs; Supernatant from LCMV ARM-infected cells was concentrated using a 10kDa centrifugal filter to yield a sample three times the concentration of unfiltered infected supernatants. The concentrated infected supernatant was then separated into filtrate and retentate using a 100kDa filter and plated onto an ELISA microplate along with the unfiltered infected supernatant and mock cell supernatant controls. Plates were then blocked and probed with purified WEN3 antibodies. Cytokine levels in TBAP cocultures were measured using Biolegend ELISA MAX Delux Set Mouse IL-2 or IL-6 respectively (Cat #431004/431304).

## Transmission electron microscopy

50 to 200 µL of concentrated infected cell supernatant was placed in a Beckman Airfuge tube and a nickel grid was inserted to the bottom of each tube. Tubes were then ultracentrifuged at 18 psi for 5 minutes at 110000g in an A-100, 45°, Airfuge rotor. (Beckman). The grids were then removed from the tubes with Dumont tweezers and dried with bibulous paper. Grids were then placed on the surface of separate drops of PBS containing 1% ovalbumin during 5 minutes at RT and then passed directly on a drop containing the primary antibody WEN3 IgG1 mouse anti LCMV GP (90min at RT). After 3 successive 5-minute washes in PBS drops, the grids were placed on the surface of separate drops of PBS containing 1% ovalbumin during 5 minutes at RT. Subsequently, grids were placed directly on drops containing the commercial secondary antibody labelled with colloidal gold (size of 5–20nm, at a dilution of 1/10) (60min at RT). Grids were washed again 3x5-minute washed in PBS and then rinsed 3x5-minutes in distilled water before being dried with bibulous paper. Lastly, grids were stained with phosphotungstic acid (PTA3%, pH6.0) and examined on transmission electron microscope (Hitachi H7100 with AMT XR111 camera).

## Statistical analysis

Prism GraphPad software was used to analyse our data, unless otherwise stated statistical significance was determined with respect to Mock using an unpaired Student's t test. *, **, and *** denote a significance of $P < 0.05$, 0.01, and 0.001, respectively.

## Supporting information

**S1 Fig. (A) CD4+T cell proliferation requires antigen recognition.** FACS plots of CFSE labeled uninfected wild-type B6 (top left panel) or SMARTA-I (top right panel) CD4+ T cells after 72 hours of co-culture with LCMV-antigen loaded wild-type B cells. Comparative histograms of CD86 expression on B cells exposed to LCMV antigen and co-cultured with

CD+ T cells from uninfected wild-type B6 (red) or SMARTA-I (blue) mice (bottom panel). (B) Filter validation experiments. FACS plots showing CFSE labelled SMARTA-I CD4+ T cells after 72 hours of co-culture with B cells loaded with either RPMI (top left panel), LCMV WE GP1 100kDa filtrate (top middle panel), or LCMV WE GP1 100kDa retentate (top right panel). Comparative histograms of CD86 expression on B cells exposed to RPMI (orange), LCMV WE GP1 filtrate (blue), or retentate (red), and co-cultured with CD4+ T cells.
(TIF)

**S2 Fig. Quantitation of viral and Defective Interfering Particle titers in LCMV mutants.** Viral titration (left) to quantify viral concentration in plaque forming units (PFU) and interference assay (right) to quantify DIP concentration in plaque interfering units50 (PIU50/ml) in wild-type LCMV (PPXY) (top) vs mutated LCMV (AAAA) (bottom) samples.
(TIF)

**S3 Fig. Expression of α-dystroglycan in various murine cells.** A western blot using the VIA4–1 antibody which recognizes glycosylated α-dystroglycan was performed on lysates from various tissues isolated from C57BL/6 (B6) mice or from murine cell lines commonly used to grow LCMV. The size of α-DG increases due to extensive O-glycosylation and variable N-glycosylation in different tissues.
(TIF)

**S4 Fig. Analysis of B cell expression of MHC-II.** B cell expression of MHC-II was measured in FACS using MFI following their co-culture with SMARTA-I T cells. B cells were incubated overnight with fractions containing low (5% and 10% Fractions) and high (Uncentrifuged and Infectious Fraction) titers of LCMV.
(TIF)

**S5 Fig. CFSE Proliferation Indexes for DIP and exosome inhibition assays.** CFSE loss for CD4+ SMARTA-I T cells was measured in FACS following co-culture with antigen exposed B cells. Proliferation Index analysis was performed using the Proliferation tool in FlowJo.
(TIF)

## Acknowledgments

We are grateful to Dr. Daniel Pinschewer of the University of Basel for generously agreeing to provide us with reagents used in this study. We also want to recognize the invaluable assistance of the staff at the LNBE animal facility as well as Jessy Tremblay and Arnaldo Nakamura who respectively operate the flow cytometry and transmission electron microscopy platforms at INRS-AFSB

## Author contributions

**Conceptualization:** Gabriel Chamberlain, Alain Lamarre.

**Data curation:** Gabriel Chamberlain, Guillaume L. Lopez, Léa Bourguignon, Rebekah Honce.

**Formal analysis:** Gabriel Chamberlain, Guillaume L. Lopez, Léa Bourguignon, Rebekah Honce.

**Funding acquisition:** Jason W Botten, Alain Lamarre.

**Investigation:** Gabriel Chamberlain, Guillaume L. Lopez, Rebekah Honce.

**Methodology:** Gabriel Chamberlain, Guillaume L. Lopez, Rebekah Honce.

**Project administration:** Gabriel Chamberlain, Jason W Botten, Alain Lamarre.

**Supervision:** Jason W Botten, Alain Lamarre.

**Writing – original draft:** Gabriel Chamberlain.

**Writing – review & editing:** Gabriel Chamberlain, Guillaume L. Lopez, Léa Bourguignon, Xavier Laulhé, Yasmine Adda-Bouchard, Tania Charpentier, Rebekah Honce, Jason W Botten, Alain Lamarre.

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
