## [Decision Letter · Decision Letter 0]

PPATHOGENS-D-25-00254

Polyclonal B cells acquire LCMV antigens in a GP-1 dependent manner

PLOS Pathogens

Dear Dr. Lamarre,

Thank you for submitting your manuscript to PLOS Pathogens. After careful consideration, we feel that it has merit and the subject matter is novel but does not fully meet PLOS Pathogens's publication criteria as it currently stands. Therefore, we invite you to submit a revised version of the manuscript that addresses the points raised during the review process, especially a major concern by the reviewer# 2 to link these studies back to hypergammaglobulinemia.  Please submit your revised manuscript within 60 days (by 6/15/25). If you will need more time than this to complete your revisions, please reply to this message or contact the journal office at plospathogens@plos.org. Please include the following items when submitting your revised manuscript:

We look forward to receiving your revised manuscript.

Kind regards,

Zia Rahman

Guest Editor

PLOS Pathogens

Matthias Schnell

Section Editor

PLOS Pathogens Sumita Bhaduri-McIntosh

Editor-in-Chief

PLOS Pathogens

orcid.org/0000-0003-2946-9497

 Michael Malim

Editor-in-Chief

PLOS Pathogens

orcid.org/0000-0002-7699-2064

**Journal Requirements:**

3) We notice that your supplementary Figures are included in the manuscript file. Please remove them and upload them with the file type 'Supporting Information'. Please ensure that each Supporting Information file has a legend listed in the manuscript after the references list.

4) We note that your Data Availability Statement is currently as follows: "All relevant data are within the manuscript and its Supporting Information files.". Please confirm at this time whether or not your submission contains all raw data required to replicate the results of your study. Authors must share the “minimal data set” for their submission. PLOS defines the minimal data set to consist of the data required to replicate all study findings reported in the article, as well as related metadata and methods (https://journals.plos.org/plosone/s/data-availability#loc-minimal-data-set-definition).

**Reviewers' Comments:**

Reviewer's Responses to Questions

**Part I - Summary**

Reviewer #1: They report that B cells are accessing LCMV viral antigen at larger than 200kda sizes in a receptor mediated manner. They show that the B cells can present the antigen even if the virus is not infectious, and this is not dependent on IDs or exosomes.

The novel finding was that B cells are acquiring antigen from articles which are larger than 200kda. This opens the door to many more questions about this process but that could be for another project

Reviewer #2: In this study, Chamberlain and colleague attempt to gain a better understanding of how B cells acquire antigens from LCMV in order to induce hypergammaglobulinemia. They use a combination of techniques and come to the conclusion that receptor mediated endocytosis is a major pathway explaining how B cells acquire antigen from LCMV. The experiments are well-controlled and generally support the conclusions. There are a few concerns the dampen the enthusiasm from the study in its current form. Most are minor concerns but the main concern is that while they are trying to link these studies back to HGG this is never tested in the current manuscript. Without establishing this link, it is this reviewer's opinion that the study would be better suited for a more specialized journal.

Reviewer #3: Chronic viral infections induce hyperglobulinemia in a T cell-dependent but B cell antigen receptor-independent manner, suggesting that viral antigens may be acquired via alternate means by polyclonal B cells. in In this study, the authors investigate novel mechanisms of antigen acquisition by polyclonal B cells using LCMV as a model system. They use T cell activation as the primary readout of antigens acquired and presented by B cells, and show that LCMV antigens are accessible to B cells in vitro from supernatants of infected cells (likely through fluid uptake, i.e., pinocytosis), and thus direct productive infection of B cells is not required. Further, the authors show that the size of antigens that are taken up by B cells is >200 kDa (likely membrane-bound vesicles but not free GP-1 monomers or trimers) as the retentates of culture supernatants passed through 100 and 200 kDa cut-off filters induced T cell activation and cytokine production but <100 and <200 kDa filtrates did not. They go on to demonstrate that antigens acquired by polyclonal B cells in this manner are capable of inducing T cell activation in vivo when such B cells are transferred into LCMV-infected mice. Finally, the authors suggest that binding of LCMV GP-1 to a potential receptor on B cells enables antigen uptake as the presence of an LCMV neutralizing antibody abrogates activation of T cells. The experiments are conducted logically and methodically. The findings open new avenues for exploring antigen uptake by polyclonal B cells in other human viral infections, and potentially identifying ways to block BCR-independent modes of uptake.

The following weaknesses must be addressed to strengthen the conclusions:

1. The authors use mechanical size separation using 100 kDa and 200 kDa cut-off filters to identify the size of antigens taken up by B cells. In both cases, retentates of the filtration system contained the antigen that is taken up and presented to T cells. Depending on the material of the filters used, is it possible that the antigens (especially free proteins) are simply adsorbed on the filters and thus not available at high enough concentration in the filtrates? Could these antigens be detected on the top surface of the filters to ensure that this is not the case?

2. In most experiments, the authors assay antigen uptake by measuring T cell proliferation, CD86 upregulation on B cells, and cytokine levels in the B-T co-cultures. What is the mechanism of CD86 upregulation in a scenario where the BCR is not involved? Are there other changes in B cell properties upon BCR-independent LCMV antigen acquisition – signaling, transcriptional activation, in vitro proliferation?

3. In Figure 5, the authors show that polyclonal B cells proliferate after presenting antigens acquired in vitro to T cells in LCMV-infected mice. It is not clear what signal these B cells are responding to.

4. As anti-LCMV antibodies block Ag presentation by polyclonal B cells in vitro, it is suggested that a GP-1 receptor may exist on B cells that mediates uptake of viral antigens (membrane vesicles?) via endocytosis. Are GP-1 or viral particles or membrane vesicles bearing viral antigens detectable inside the B cell endosomes? There are ways to block endocytosis and should be used to test if this is the only mechanism or one of the mechanisms by which polyclonal B cells acquire antigen in the in vitro system utilized in this study. This is important as pinocytosis is also considered as a potential mechanism of antigen uptake.

**Part II – Major Issues: Key Experiments Required for Acceptance**

Reviewer #1: None. They experimentally address the questions that they ask.

Reviewer #2: The authors never show that blocking receptor mediated endocytosis lead to any impact on HGG which the major rationale for these studies. Without this link, the enthusiasm for this study is significantly dampened

Reviewer #3: The following concerns must be addressed through experimentation and discussion:

1. The authors use mechanical size separation using 100 kDa and 200 kDa cut-off filters to identify the size of antigens taken up by B cells. In both cases, retentates of the filtration system contained the antigen that is taken up and presented to T cells. Depending on the material of the filters used, is it possible that the antigens (especially free proteins) are simply adsorbed on the filters and thus not available at high enough concentration in the filtrates? Could these antigens be detected on the top surface of the filters to ensure that this is not the case?

2. In most experiments, the authors assay antigen uptake by measuring T cell proliferation, CD86 upregulation on B cells, and cytokine levels in the B-T co-cultures. What is the mechanism of CD86 upregulation in a scenario where the BCR is not involved? Are there other changes in B cell properties upon BCR-independent LCMV antigen acquisition – signaling, transcriptional activation, in vitro proliferation?

3. As anti-LCMV antibodies block Ag presentation by polyclonal B cells in vitro, it is suggested that a GP-1 receptor may exist on B cells that mediates uptake of viral antigens (membrane vesicles?) via endocytosis. Are GP-1 or viral particles or membrane vesicles bearing viral antigens detectable inside the B cell endosomes? There are ways to block endocytosis and should be used to test if this is the only mechanism or one of the mechanisms by which polyclonal B cells acquire antigen in the in vitro system utilized in this study. This is important as pinocytosis is also considered as a potential mechanism of antigen uptake.

**Part III – Minor Issues: Editorial and Data Presentation Modifications**

Reviewer #1: Figure 1. Should the statistical analysis be unpaired T test when comparing more than 2 variables? You may get the same answer, but the ANOVA and posttest would be a sounder statistical analysis.

Figure 6. Because the sups from WT LCMV infected cells had a 50-fold higher titer than the LCMV-AAAA, you diluted it to control for infectious content but does your dilution now compare a very low DI level to an undetectable DI level? Thus, giving the results observed?

Figure 7. Is there a reason for switching to the Cl-13 strain in this experiment?

Reviewer #2: In figure 2 it appears that the majority of cells have divided at least once but this is ignored and not discussed. This is either due to some baseline proliferation or poor CFSE staining but this needs to be addressed.

The data on proliferation is only analyzed based on the percentage of cells that have divided but this is problematic. Measuring a proliferation index that takes into account how many times the cells have divided would be much more informative and would likely detect some of the more minor differences. For example, in figure 6A it appears that while a similar percentages of cells have initiated division, more cells have undergone extensive division in the PPXY group. Since some of the cells that have completely diluted CFSE may have undergone several more rounds of division, these data are important. This analysis should be provided throughout the study.

Reviewer #3: (No Response)

PLOS authors have the option to publish the peer review history of their article (what does this mean? ). If published, this will include your full peer review and any attached files.

**Do you want your identity to be public for this peer review?** For information about this choice, including consent withdrawal, please see our Privacy Policy .

Reviewer #1: No

Reviewer #2: No

Reviewer #3: No

**Figure resubmission:**
---

## [Decision Letter · Decision Letter 1]

Dear Dr. Lamarre,

We are pleased to inform you that your manuscript 'Polyclonal B cells acquire LCMV antigens in a GP1-dependent manner' has been provisionally accepted for publication in PLOS Pathogens.

Best regards,

Zia Rahman

Guest Editor

PLOS Pathogens

Matthias Schnell

Section Editor

PLOS Pathogens

Sumita Bhaduri-McIntosh

Editor-in-Chief

PLOS Pathogens

orcid.org/0000-0003-2946-9497

Michael Malim

Editor-in-Chief

PLOS Pathogens

orcid.org/0000-0002-7699-2064

Reviewer Comments (if any, and for reference):

Reviewer's Responses to Questions

**Part I - Summary**

Reviewer #1: (No Response)

Reviewer #2: In this revised manuscript the authors addressed some of the reviewer concerns but in our opinion still fell short of addressing the major concerns since the novelty of the study really lies in the mechanism through which the cells acquire antigen. In this revision they have been unable to clearly establish this mechanism beyond correlations and to clearly link their findings to HGG. As such it is still this reviewer's opinion that this study is more suited for a specialized journal.

Reviewer #3: The authors have satisfactorily addressed all comments from previous review and provided substantial amount of new data to support their interpretation and conclusions that polyclonal B cells recognize GP1 on LCMV through a receptor-mediated mechanism.

**Part II – Major Issues: Key Experiments Required for Acceptance**

Reviewer #1: (No Response)

Reviewer #2: (No Response)

Reviewer #3: No major issues were identified in the revised manuscript.

**Part III – Minor Issues: Editorial and Data Presentation Modifications**

Reviewer #1: (No Response)

Reviewer #2: (No Response)

Reviewer #3: No minor issues were identified in the revised manuscript.

PLOS authors have the option to publish the peer review history of their article (what does this mean? ). If published, this will include your full peer review and any attached files.

**Do you want your identity to be public for this peer review?** For information about this choice, including consent withdrawal, please see our Privacy Policy .

Reviewer #1: No

Reviewer #2: No

Reviewer #3: No

---

## [Editor Report · Acceptance letter]

Dear Prof Lamarre,

We are delighted to inform you that your manuscript, "Polyclonal B cells acquire LCMV antigens in a GP1-dependent manner," has been formally accepted for publication in PLOS Pathogens.

Best regards,

Sumita Bhaduri-McIntosh

Editor-in-Chief

PLOS Pathogens

orcid.org/0000-0003-2946-9497

Michael Malim

Editor-in-Chief

PLOS Pathogens

orcid.org/0000-0002-7699-2064